# Pseudo Label-Guided Multi Task Learning for Scene Understanding

## Abstract

Multi-task learning (MTL) for scene understanding has been actively studied by exploiting correlation of multiple tasks. This work focuses on improving the performance of the MTL network that infers depth and semantic segmentation maps from a single image. Specifically, we propose a novel MTL architecture, called Pseudo-MTL, that introduces pseudo labels for joint learning of monocular depth estimation and semantic segmentation tasks. The pseudo ground truth depth maps, generated from pretrained stereo matching methods, are leveraged to supervise the monocular depth estimation. More importantly, the pseudo depth labels serve to impose a *cross-view* consistency on the estimated monocular depth and segmentation maps of two views. This enables for mitigating the mismatch problem incurred by inconsistent prediction results across two views. A thorough ablation study validates that the cross-view consistency leads to a substantial performance gain by ensuring inference-view invariance for the two tasks.

## 1 Introduction

Scene understanding has become increasingly popular in both academia and industry as an essential technology for realizing a variety of vision-based applications such as robotics and autonomous driving. 3D geometric and semantic information of a scene often serve as a basic building block for high-level scene understanding tasks. Numerous approaches have been proposed for inferring a depth map (Garg et al., 2016; Godard et al., 2019) or grouping semantically similar parts (Chen et al., 2017; Yuan et al., 2019) from a single image. In parallel with such a rapid evolution for individual tasks, several approaches (Chen et al., 2019; Zhang et al., 2018; Guizilini et al., 2020b; Liu et al., 2019) have focused on boosting the performance through joint learning of the semantic segmentation and monocular depth estimation tasks by considering that the two tasks are highly correlated. For instance, pixels with the same semantic segmentation labels within an object are likely to have similar (or smoothly-varying) depth values. An abrupt change of depth values often implies the boundary of two objects containing different semantic segmentation labels. These properties have been applied to deep networks to enhance the semantic segmentation and monocular depth estimation tasks in a synergetic manner.

In (Chen et al., 2019), they proposed a joint learning model that learns semantic-aware representation to advance the monocular depth estimation with the aid of semantic segmentation. A depth map is advanced by making use of loss functions designed for the purpose of bonding geometric and semantic understanding. The method in (Guizilini et al., 2020b) proposed a new architecture that improves the accuracy of monocular depth estimation through the pixel-adaptive convolution (Su et al., 2019) using semantic feature maps computed from pre-trained semantic segmentation networks. Despite the improved monocular depth accuracy over a single monocular depth network, the performance improvement of the semantic segmentation task by the aid of geometrical representation has not been verified (Chen et al., 2019), or even the semantic segmentation network was fixed with pretrained parameters (Guizilini et al., 2020b).

A generic computational approach for multi-task learning (MTL) was proposed in (Zamir et al., 2018), which models the structure across twenty six tasks, including 2D, 2.5D, 3D, and semantic tasks, by finding first and higher-order transfer learning dependencies across them in a latent space to seamlessly reuse supervision among related tasks and/or solve them in a single network without increasing the complexity significantly. This was further extended by imposing a cross-task consis-

tency based on inference-path invariance on a graph of multiple tasks (Zamir et al., 2020). Though these approaches provide a generic and principled way for leveraging redundancies across multiple tasks, there may be limitations to improving the performance of individual tasks in that it is difficult to consider task-specific architectures and loss functions in such unified frameworks. With the same objective yet with a different methodology, the method in (Liu et al., 2019) proposes a novel MTL architecture consisting of task-shared and task-specific networks based on task-attention modules, aiming to learn both generalizable features for multiple tasks and features tailored to each task. They validated the performance in the joint learning of monocular depth and semantic segmentation.

In this paper, we propose a novel MTL architecture for monocular depth estimation and semantic segmentation tasks, called pseudo label-guided multi-task learning (Pseudo-MTL). The proposed architecture leverages geometrically- and semantically-guided representations by introducing pseudo ground truth labels. When a pair of stereo images is given as inputs, our method first generates *pseudo* ground truth left and right depth maps by using existing pre-trained stereo matching networks (Pang et al., 2017; Chang & Chen, 2018). To prevent inaccurate depth values from being used, a stereo confidence map (Poggi & Mattoccia, 2016) is used together as auxiliary data that measures the reliability of the pseudo depth labels. These are leveraged for supervising the monocular depth network, obtaining substantial performance gain over recent self-supervised monocular depth estimation approaches (Godard et al., 2017; 2019). More importantly, the pseudo depth labels are particularly useful when imposing a *cross-view* consistency across left and right images. The estimated monocular depth and segmentation maps of two views are tied from a geometric perspective by minimizing the cross-view consistency loss, alleviating the mismatch problem incurred by inconsistent prediction across two views significantly. We will verify through an intensive ablation study that the proposed cross-consistency loss leads to a substantial improvement on both tasks. Experimental results also show that our approach achieves an outstanding performance over state-of-the-arts. In short, our novel contributions can be summarized as follows.

- We propose a novel MTL approach that jointly performs monocular depth estimation and semantic segmentation through pseudo depth labels.
- The cross-view consistency loss based on the pseudo depth labels and associated confidence maps is proposed to enable consistent predictions across two views.
- An intensive ablation study is provided to quantify the contribution of the proposed items to performance improvement.

## 2 RELATED WORK

**Monocular Depth Estimation** While early works for monocular depth estimation are based on supervised learning, self-supervised learning has attracted increasing interest in recent approaches (Godard et al., 2017; 2019; Watson et al., 2019) to overcome the lack of ground truth depth labels. Here, we review works mostly relevant to our method. Godard et al. (Godard et al., 2017; 2019) proposed the deep network that infers a disparity map using the image reconstruction loss and left-right consistency loss from a pair of stereo images or monocular videos. Chen et al. (Chen et al., 2019) infers both disparity and semantic segmentation maps by enforcing the cross consistency across stereo images to address the mismatch problem of (Godard et al., 2017). Several approaches have focused on improving the monocular depth estimation through the aid of segmentation networks, e.g., by stitching local depth segments from instance segmentation with respect to scale and shift (Wang et al., 2020) or leveraging pretrained semantic segmentation networks to guide the monocular depth estimation (Guizilini et al., 2020b).

**Semantic Segmentation** A deep convolutional encoder-decoder architecture for semantic segmentation proposed in (Badrinarayanan et al., 2017) has been widely used as backbone. The pyramid pooling module was proposed for leveraging global context through aggregation of different region-based contexts (Zhao et al., 2017). Some segmentation works attempted to combine different tasks to improve segmentation performance. Gated-SCNN (Takikawa et al., 2019) refines segmentation results by fusing semantic-region features and boundary features. FuseNet (Hazirbas et al., 2016) proposed to fuse features from color and depth images for improving the segmentation performance.

**Multi-task learning** In (Chen et al., 2019; Takikawa et al., 2019; Zhang et al., 2018), they proposed to leverage task-specific loss functions to tie up two (or more) tasks within the MTL architecture. For

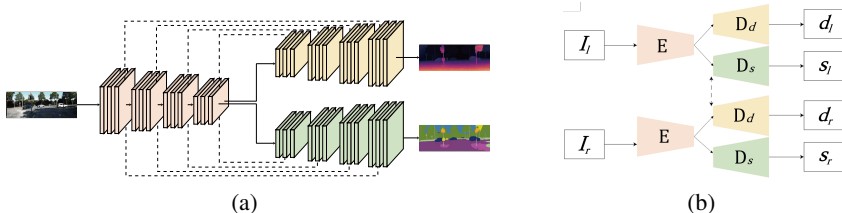

(a)                             (b)

Figure 1: Network architecture: (a) Pseudo MTL based on the encoder-decoder, (b) To impose the cross-view consistency, it is applied for left and right images, respectively.

instance, Chen et al. (Chen et al., 2019) attempted to improve a monocular depth accuracy by using the loss functions that measure the consistency between geometric and semantic predictions. The generic computational approach for MTL was proposed by leveraging redundancies across multiple tasks in a latent space in (Zamir et al., 2018; 2020). The task-attention modules were introduced to extract features for individual tasks in (Misra et al., 2016; Liu et al., 2019; Jha et al., 2020). In this work, we focus on improving the performance of the MTL architecture for monocular depth estimation and semantic segmentation tasks by using the cross-view consistency loss based on pseudo labels.

## 3   Proposed Method

### 3.1   Overview and Architecture Design

Our Pseudo-MTL approach focuses on improving the performance of the monocular depth estimation and semantic segmentation tasks through task-specific losses defined based on the *pseudo* depth labels generated by using pre-trained stereo matching networks (Pang et al., 2017). The stereo confidence maps are used together as auxiliary data to compensate for estimation errors in the pseudo depth labels. These are effective in mitigating undesired artifacts of errors that may exist in the pseudo depth labels. In our work, we chose the CCNN (Poggi & Mattoccia, 2016) for calculating the confidence map, but more advanced confidence estimation approaches can also be used.

As shown in Figure 1, the proposed Pseudo-MTL network is based on the encoder-decoder architecture, in which a single encoder takes an image and two decoders predict the monocular depth and semantic segmentation maps. The encoder network E consists of the convolutional layers of the VGG network (Simonyan & Zisserman, 2015). Two decoders, $D_d$ for monocular depth estimation and $D_s$ for monocular depth estimation, are designed symmetrically with the encoder. While two tasks share the encoder, the task-specific decoder branches are used for each task.

The pseudo depth labels and the segmentation label maps of stereo images are used for supervising the proposed architecture. The monocular depth and segmentation maps of left and right images are estimated by passing each image to the proposed architecture, as shown in Figure 1. The cross-view consistency loss is then imposed on the prediction results of two views. To be specific, the estimated monocular depth maps of left and right images are warped and tested using the pseudo depth labels for ensuring inference-view invariance on the monocular depth estimation, and a similar procedure is also applied to the semantic segmentation.

Using the pseudo depth labels for training the proposed model is advantageous at various aspects. The pseudo depth labels of stereo images, filtered out by its confidence map, provides a better supervision (Choi et al., 2020) than recent self-supervised monocular depth estimation approaches. More importantly, the cross-view consistency based on the pseudo depth labels mitigates the mismatch problem by inconsistent prediction results of two views, leading to a substantial performance gain. Our method aims at advancing the two tasks via task-specific losses based on pseudo ground truth labels, and existing MTL architectures, e.g. based on task-specific attention modules and adaptive balancing (Liu et al., 2019; Jha et al., 2020), can be used complementarily with our loss functions.

### 3.2   Loss Functions

Loss functions are divided into two parts, 1) supervised loss for depth and segmentation networks and 2) pseudo depth-guided reconstruction loss for cross-view consistency. Note that the supervised

loss used for monocular depth estimation relies on the pseudo depth labels generated from a pair of stereo images.

### 3.2.1 LOSS FOR MONOCULAR DEPTH AND SEMANTIC SEGMENTATION

Depth maps $d_i$ for $i = \{l, r\}$, predicted by the decoder $\mathtt{D}_d$ for monocular depth estimation, are used for measuring the depth regression loss $L_d$ as follows:

$$
L_d = \sum_{i=\{l,r\}} L_{reg}(c_i, d_i, d_i^{\mathrm{pgt}}), \quad \text{where} \ \ L_{reg}(c_i, d_i, d_i^{\mathrm{pgt}}) = \frac{1}{Z_i} \sum_{p \in \Phi} c_i(p) \cdot |d_i(p) - d_i^{\mathrm{pgt}}(p)|_1,
\tag{1}
$$

where $c_i$ and $d_i^{\mathrm{pgt}}$ indicate the confidence map and pseudo ground truth depth map of left ($i = l$) or right ($i = r$) images, respectively. The loss is normalized with $Z_i = \sum_p c_i(p)$. $\Phi$ represents a set of all pixels. The confidence map serves to exclude inaccurate depth values of $d_i^{\mathrm{pgt}}$ when calculating the depth regression loss $L_d$. This can be used in various ways, including the hard thresholding (Cho et al., 2019; Tonioni et al., 2020) and the soft thresholding (Choi et al., 2020). Among them, the soft thresholded confidence map (Choi et al., 2020) is shown to be effective in the monocular depth estimation. Our work chose to threshold the confidence map through the soft-thresholding of (Choi et al., 2020). We found that the pretrained threshold network already provides satisfactory results, and thus it was fixed during our network training.

A supervised loss for semantic segmentation is defined with the standard cross-entropy $H$:

$$
L_s = \sum_{i=\{l,r\}} H(s_i, s_i^{\mathrm{gt}}),
\tag{2}
$$

$s_i$ and $s_i^{\mathrm{gt}}$ denote the segmentation map, predicted by the decoder $\mathtt{D}_s$ for semantic segmentation, and ground truth segmentation map, respectively. The supervised loss for both tasks is defined as $L_S = \alpha_d L_d + \alpha_s L_s$ with loss weights $\alpha_d$ and $\alpha_s$.

### 3.2.2 CROSS-VIEW CONSISTENCY LOSS

Minimizing the supervised loss $L_S$ for an individual view may often lead to the mismatched problem in the predicted depth and segmentation maps due to the lack of consistency constraints across two views. We address this issue by imposing the cross-view consistency across left and right images with the pseudo depth labels. Figure 2 shows the procedure of computing the cross-view consistency losses with pseudo depth labels. The cross-view consistency loss for the monocular depth estimation is defined as follows:

$$
L_{d,c} = \alpha_{d,lr} L_{d,lr} + \alpha_{d,l} L_{d,l} + \alpha_{d,r} L_{d,r},
\tag{3}
$$

$$
L_{d,lr} = L_{reg}\left(c_l, d_l, G(d_r; d_l^{\mathrm{pgt}})\right) + L_{reg}\left(c_r, G(d_l; d_r^{\mathrm{pgt}}), d_r\right),
\tag{4}
$$

$$
L_{d,l} = L_{reg}\left(c_l, d_l^{\mathrm{pgt}}, G(d_r; d_l^{\mathrm{pgt}})\right) + L_{reg}\left(c_l, d_l, G(d_r^{\mathrm{pgt}}; d_l^{\mathrm{pgt}})\right),
\tag{5}
$$

$$
L_{d,r} = L_{reg}\left(c_r, G(d_l; d_r^{\mathrm{pgt}}), d_r^{\mathrm{pgt}}\right) + L_{reg}\left(c_r, G(d_l^{\mathrm{pgt}}; d_r^{\mathrm{pgt}}), d_r\right),
\tag{6}
$$

where $\alpha_{d,lr}$, $\alpha_{d,l}$, and $\alpha_{d,r}$ denote weights for each loss. $G(a; b)$ indicates the result of warping $a$ with a depth map $b$ into another view. For instance, $G(d_r; d_l^{\mathrm{pgt}})$ returns the depth map warped onto the left image using $d_l^{\mathrm{pgt}}$. $L_{d,lr}$ measures the cross-view consistency between two predicted depth maps $d_l$ and $d_r$. Note that the warping function $G$ is applied to $d_r$ and $d_l$, respectively. Similar to the depth regression loss $L_d$, the confidence map is used together to prevent inaccurate values in the pseudo depth labels from being used. $L_{d,l}$ denotes the cross-view consistency for ($d_l^{\mathrm{pgt}}$, $d_r$) and ($d_l$, $d_r^{\mathrm{pgt}}$) using the left pseudo label $d_l^{\mathrm{pgt}}$. This implies that when warping $d_r$ (or $d_r^{\mathrm{pgt}}$) into the left image, the warped result should be similar to $d_l^{\mathrm{pgt}}$ (or $d_l$). $L_{d,r}$ is defined in a similar manner.

The cross-view consistency can also be applied to semantic segmentation as follows:

$$L_{s,c} = \alpha_{s,lr} L_{s,lr} + \alpha_{s,l} L_{s,l} + \alpha_{s,r} L_{s,r}, \tag{7}$$

$$L_{s,lr} = c_l \cdot H\left(s_l, G(s_r; d_l^{\mathrm{pgt}})\right) + c_r \cdot H\left(G(s_l; d_r^{\mathrm{pgt}}), s_r\right), \tag{8}$$

$$L_{s,l} = c_l \cdot H\left(s_l^{\mathrm{gt}}, G(s_r; d_l^{\mathrm{pgt}})\right) + c_l \cdot H\left(s_l, G(s_r^{\mathrm{gt}}; d_l^{\mathrm{pgt}})\right), \tag{9}$$

$$L_{s,r} = c_r \cdot H\left(G(s_l; d_r^{\mathrm{pgt}}), s_r^{\mathrm{gt}}\right) + c_r \cdot H\left(G(s_l^{\mathrm{gt}}; d_r^{\mathrm{pgt}}), s_r\right), \tag{10}$$

where '$\cdot$' indicates an element-wise multiplication. The confidence maps $c_l$ and $c_r$ are also used to compensate for errors in the pseudo depth labels $d_l^{\mathrm{pgt}}$ and $d_r^{\mathrm{pgt}}$. Note that for some training datasets that provide no ground truth segmentation maps, we generate pseudo ground truth segmentation maps. More details are provided in Section 3.3.

Note that in (Chen et al., 2019), the consistency for left and right segmentation maps is considered e.g., by minimizing $H(s_l, G(s_r; d_l))$. Two segmentation maps $s_l$ and $s_r$ are aligned with the estimated monocular depth map $d_l$. However, $d_l$ is continuously updated during the network training, and thus this may result in inaccurate alignments at early stage, often leading to divergences of loss. For these reasons, minimizing the loss $H$ with respect to both monocular depth and segmentation maps often becomes very challenging, and the performance gain by the consistency loss is relatively marginal. Contrarily, our approach is more effective in imposing the cross-view consistency in that 1) more accurate pseudo depth labels, obtained from stereo matching networks, are used, 2) the confidence map helps to filter out inaccurate depth values in the pseudo ground truth depth maps. Furthermore, we extend the cross-view consistency to the monocular depth estimation, which

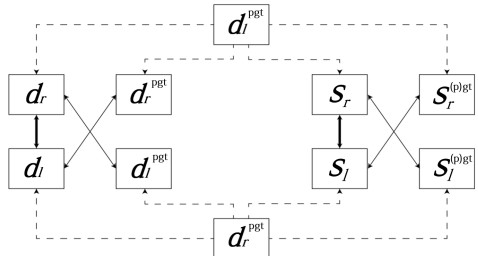

Figure 2: Cross-view consistency loss. Monocular depth or semantic segmentation maps are warped using $d_l^{\mathrm{pgt}}$ and $d_r^{\mathrm{pgt}}$, and the consistency losses are measured using equation 3 and equation 7. Dotted lines mean a warping operator $G(a; b)$, and solid lines denote the cross-view consistency losses. As summarized in Table 1, either $s_i^{pgt}$ or $s_i^{gt}$ is used as supervision for semantic segmentation.

is infeasible in the recent self-supervised monocular depth estimation approaches (Godard et al., 2017; 2019; Watson et al., 2019) that rely on the reconstruction loss only. A detailed ablation study will be provided to validate the effectiveness of the proposed cross-view consistency loss. A total loss is defined as

$$L = L_S + L_{d,c} + L_{s,c}. \tag{11}$$

## 3.3 TRAINING DETAILS

While the pseudo depth labels $d_l^{\mathrm{pgt}}$ and $d_r^{\mathrm{pgt}}$, generated using pretrained stereo matching networks, are used to supervise the monocular depth estimation task, the semantic segmentation task requires using the ground truth segmentation maps. The Cityscapes dataset provide only the left ground truth segmentation map $s_l^{\mathrm{gt}}$, and the KITTI dataset does not provide them. In our work, we hence generated the pseudo segmentation labels of these images by using semantic segmentation methods (Cheng et al., 2020; Zhu et al., 2019). Table 1 summarizes the supervisions used for the two tasks.

Table 1: Supervision used in the KITTI and Cityscapes datasets.

| Task | Input | KITTI | Cityscapes |
|---|---|---|---|
| Seg. | $I_l$ | $s_l^{\mathrm{pgt}}$ | $s_l^{\mathrm{gt}}$ |
| | $I_r$ | $s_r^{\mathrm{pgt}}$ | $s_r^{\mathrm{pgt}}$ |
| Depth | $I_l$ | $d_l^{\mathrm{pgt}}$ | $d_l^{\mathrm{pgt}}$ |
| | $I_r$ | $d_r^{\mathrm{pgt}}$ | $d_r^{\mathrm{pgt}}$ |

Table 2: Quantitative evaluation of monocular depth estimation on Eigen split of KITTI dataset. Numbers in bold and underlined represent $1^{st}$ and $2^{nd}$ ranking, respectively. The methods used in evaluation are (Garg et al., 2016), (Zhou et al., 2017), Monodepth (Godard et al., 2017), (Zhan et al., 2018), (Chen et al., 2019), Monodepth2 (Godard et al., 2019), Uncertainty (Poggi et al., 2020), DepthHint (Watson et al., 2019), (Guizilini et al., 2020b), and (Choi et al., 2020).

| Method | cap | (Lower is better) | | | | (Higher is better) | | |
|---|---|---|---|---|---|---|---|---|
| | | Abs Rel | Sq Rel | RMSE | RMSE log | $\delta < 1.25$ | $\delta < 1.25^2$ | $\delta < 1.25^3$ |
| Garg et al. | | 0.152 | 1.226 | 5.849 | 0.246 | 0.784 | 0.921 | 0.967 |
| Zhou et al. | | 0.150 | 1.124 | 5.507 | 0.223 | 0.806 | 0.933 | 0.973 |
| Monodepth | | 0.138 | 1.186 | 5.650 | 0.234 | 0.813 | 0.930 | 0.969 |
| Zhan et al. | | 0.135 | 1.132 | 5.585 | 0.229 | 0.820 | 0.933 | 0.971 |
| Chen et al. | 80m | 0.118 | 0.905 | 5.096 | 0.211 | 0.839 | 0.945 | 0.977 |
| Monodepth2 | | 0.108 | 0.842 | 4.891 | 0.207 | 0.866 | 0.949 | 0.976 |
| Uncertainty | | 0.107 | 0.811 | 4.796 | 0.200 | 0.866 | 0.952 | 0.978 |
| DepthHint | | 0.102 | 0.762 | 4.602 | 0.189 | 0.880 | 0.960 | 0.981 |
| Guizilini et al. | | 0.102 | 0.698 | 4.381 | **0.178** | **0.896** | **0.964** | **0.984** |
| Choi et al. | | 0.098 | 0.647 | 4.253 | 0.186 | 0.884 | 0.960 | 0.981 |
| **Ours** | | **0.097** | **0.599** | **4.197** | 0.184 | 0.883 | 0.962 | 0.982 |

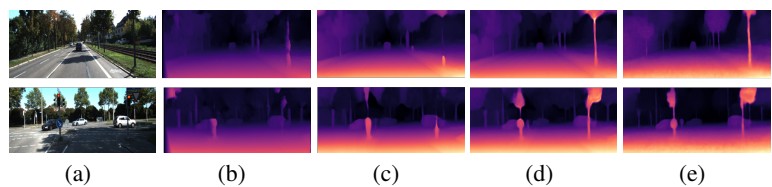

(a)    (b)    (c)    (d)    (e)

Figure 3: Qualitative evaluation of monocular depth estimation on Eigen split of KITTI dataset. (a) Input image, (b) Monodepth (Godard et al., 2017), (c) Monodepth2 (Godard et al., 2019), (d) DepthHints (Watson et al., 2019), and (e) Ours.

## 4 EXPERIMENTAL RESULTS

### 4.1 DATASETS

We evaluated the performance on two popular datasets, KITTI (Geiger et al., 2012) and Cityscapes (Cordts et al., 2016). In KITTI, for a fair comparison, we followed the common setup to use 22,600 images for training and the rest for evaluation. The Eigen split data (697 images) (Eigen et al., 2014) was used for evaluating the monocular depth accuracy. Following existing MTL methods (Chen et al., 2019), the semantic segmentation accuracy was evaluated with 200 annotated data provided from KITTI benchmark. Cityscapes provides high resolution images of urban street scenes used for segmentation and depth estimation. 2,975 and 500 images were used for training and evaluation, respectively.

### 4.2 IMPLEMENTATION DETAILS AND EVALUATION METRIC

We first pretrained the monocular depth network $E + D_d$ and semantic segmentation network $E + D_s$ independently for 30 epochs using the Adam optimizer (Kingma & Ba, 2015) with a learning rate of $10^{-4}$ and momentum of 0.9. We then finetuned the whole network $E + D_d + D_s$ for 20 epochs using the Adam optimizer with a learning rate of $10^{-5}$, reduced to 1/10 every 10 epochs, and a momentum of 0.9, after initializing it with the pretrained weight parameters of the monocular depth network $E + D_d$ and semantic segmentation network $D_s$. During training, we resized KITTI images to a resolution of [480, 192], and cropped Cityscapes images [2048, 768] to exclude the front part of the car and resized to a resolution of [256, 96]. The weights for the objective function are set to $\alpha_d$ = 850, $\alpha_s$ = 2.5, $\alpha_{d,lr}$ = 0.5, $\alpha_{d,l}$ = 1 ,$\alpha_{d,r}$ = 1, $\alpha_{s,lr}$ = 0.5, $\alpha_{s,l}$ = 1.5, $\alpha_{s,r}$ = 1.5. The performance evaluation was conducted by following the common practices: 1) mean absolute relative error (Abs Rel), mean relative squared error (Sq Rel), root mean square error (RMSE), root mean square error log (RMSE log) and accuracy under threshold $\delta$ for monocular depth estimation, 2) intersection over union (IoU) and mean intersection over union (mIoU) for semantic segmentation. Due to page limits, some results are provided in appendix. Our code will be publicly available later.

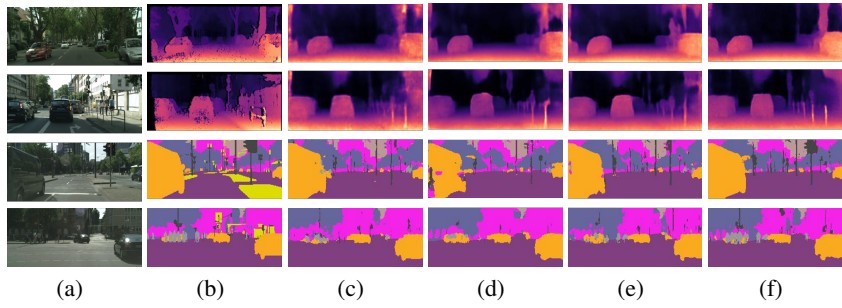

Figure 4: Qualitative results of semantic segmentation prediction on KITTI dataset.

(a)          (b)          (c)          (d)          (e)          (f)

Figure 5: Qualitative results of monocular depth estimation on the Cityscapes dataset: (a) Input image, (b) Ground truth, (c) Cross-stitch (Misra et al., 2016), (d) MTAN (Liu et al., 2019), (e) Dense (Liu et al., 2019), and (f) Ours. Note that ground truth depth maps were obtained using SGM (Hirschmuller, 2008).

## 4.3 PERFORMANCE EVALUATION

**KITTI** In Table 2, we provide objective evaluation results on the KITTI Eigen split (Eigen et al., 2014). The proposed method produces very competitive results to state-of-the-arts monocular depth estimation approaches. Qualitative evaluation in Figure 3 verifies that our method yields the results with sharper boundaries and better object delineation. These validate the effectiveness of the cross-view consistency based on the pseudo depth labels. In Figure 4, the proposed method produces satisfactory semantic segmentation results for the Cityscapes dataset, achieving mIoU = 59.93. Note that mIoU in the MTL approach of (Chen et al., 2019) is 39.13.

**Cityscapes** In Table 3, we compared results on the Cityscape dataset with recent multi-task learning approaches for monocular depth estimation and semantic segmentation tasks: 'Cross-stitch' (Misra et al., 2016) and 'MTAN' (Liu et al., 2019). 'Split (deep)', 'Split (wide)', and 'Dense' were reproduced by using author-provided codes in 'MTAN' (Liu et al., 2019). Our method achieves improved quantitative results on both tasks. Figure 5 exhibits qualitative results on the Cityscape dataset. As expected, depth and segmentation maps generated by our method are capable of preserving object boundaries and recover details better than the latest MTL methods (Misra et al., 2016; Liu et al., 2019).

Table 3: Multi-task validation results for 7-class semantic segmentation and depth estimation on Cityscapes dataset.

| Method | Segmentation (Higher is better) | | Depth (Lower is better) | |
|---|---|---|---|---|
| | mIoU | IoU | Abs | Sq |
| Split (deep) | 56.24 | 88.13 | 0.644 | 0.259 |
| Cross-stitch | 59.77 | 89.70 | 0.610 | 0.251 |
| Split (wide) | 59.71 | 89.84 | 0.619 | 0.254 |
| MTAN | 51.08 | 89.77 | 0.669 | 0.250 |
| Dense | 60.01 | 89.81 | 0.598 | 0.242 |
| Ours | 64.76 | 91.90 | 0.542 | 0.228 |

## 4.4 ABLATION STUDY

We conducted the ablation experiments to validate the effectiveness of the confidence map and cross-view consistency for the KITTI dataset in Table 4 and the Cityscapes dataset in Table 5. We first compared the performance with the method ($b = d_i$) based on the cross-view consistency using the estimated monocular depth map, e.g. $H(s_l, G(s_r; d_l))$, similar to (Chen et al., 2019). Under the same setting, our method ($b = d_i^{\text{pgt}}$) tends to achieve higher mIoU than the method ($b = d_i$). Ad-

Table 4: Ablation study of our model on the KITTI dataset. 'Baseline' model is our network without the confidence and cross consistency loss.

| Method | $G(-;b)$ | Proposed components | | | | Depth | | | | | | Seg |
|---|---|---|---|---|---|---|---|---|---|---|---|---|
| | | | | | | (Lower is better) | | (Higher is better) | | | | |
| | | | | | | Rel | RMSE | | $\delta$ | | | |
| | | $c_i$ | $L_{s,lr}$ | $L_{s,l}/L_{s,r}$ | $L_{d,c}$ | Abs / Sq | raw / log | 1.25 | $1.25^2$ | $1.25^3$ | | mIoU |
| Baseline | $b = d_i^{\mathrm{pgt}}$ | | | | | 0.103 / 0.673 | 4.500 / 0.194 | 0.871 | 0.957 | 0.980 | | 56.90 |
| Ours | $b = d_i$ | | ✓ | | | 0.101 / 0.665 | 4.510 / 0.191 | 0.873 | 0.958 | 0.981 | | 58.26 |
| | $b = d_i$ | | | ✓ | | 0.103 / 0.674 | 4.537 / 0.196 | 0.869 | 0.956 | 0.980 | | 58.44 |
| | $b = d_i$ | | ✓ | ✓ | | 0.104 / 0.678 | 4.579 / 0.196 | 0.867 | 0.955 | 0.979 | | 59.00 |
| | $b = d_i^{\mathrm{pgt}}$ | | ✓ | | | 0.100 / 0.661 | 4.461 / 0.191 | 0.876 | 0.959 | 0.981 | | 58.51 |
| | $b = d_i^{\mathrm{pgt}}$ | | | ✓ | | 0.100 / 0.660 | 4.471 / 0.191 | 0.876 | 0.958 | 0.981 | | 59.16 |
| | $b = d_i^{\mathrm{pgt}}$ | | ✓ | ✓ | | 0.101 / 0.668 | 4.518 / 0.194 | 0.872 | 0.958 | 0.980 | | 59.41 |
| | $b = d_i^{\mathrm{pgt}}$ | ✓ | | | | 0.099 / 0.611 | 4.268 / 0.186 | 0.882 | 0.962 | 0.982 | | 56.59 |
| | $b = d_i^{\mathrm{pgt}}$ | ✓ | ✓ | | | 0.096 / 0.612 | 4.285 / 0.185 | 0.884 | 0.962 | 0.982 | | 58.91 |
| | $b = d_i^{\mathrm{pgt}}$ | ✓ | | ✓ | | 0.097 / 0.610 | 4.282 / 0.183 | 0.884 | 0.962 | 0.983 | | 59.78 |
| | $b = d_i^{\mathrm{pgt}}$ | ✓ | ✓ | ✓ | | 0.096 / 0.616 | 4.287 / 0.184 | 0.884 | 0.962 | 0.982 | | 60.21 |
| | $b = d_i^{\mathrm{pgt}}$ | ✓ | | | ✓ | 0.100 / 0.613 | 4.213 / 0.186 | 0.878 | 0.961 | 0.982 | | 56.79 |
| | $b = d_i^{\mathrm{pgt}}$ | ✓ | ✓ | ✓ | ✓ | 0.097 / 0.599 | 4.197 / 0.184 | 0.883 | 0.962 | 0.982 | | 59.93 |

Table 5: Ablation study of our model on the Cityscapes dataset. 'Baseline' model is our network without the confidence and cross consistency loss.

| Method | $G(-;b)$ | Proposed components | | | | Depth | | Seg | |
|---|---|---|---|---|---|---|---|---|---|
| | | | | | | (Lower is better) | | (Higher is better) | |
| | | $c_i$ | $L_{s,lr}$ | $L_{s,l}/L_{s,r}$ | $L_{d,c}$ | Abs | Sq | mIoU | Pixel Acc |
| Baseline | $b = d_i^{\mathrm{pgt}}$ | | | | | 0.584 | 0.246 | 63.01 | 91.24 |
| Ours | $b = d_i$ | | ✓ | | | 0.573 | 0.238 | 63.83 | 91.53 |
| | $b = d_i$ | | | ✓ | | 0.586 | 0.244 | 64.05 | 91.54 |
| | $b = d_i$ | | ✓ | ✓ | | 0.584 | 0.243 | 64.15 | 91.59 |
| | $b = d_i^{\mathrm{pgt}}$ | | ✓ | | | 0.572 | 0.242 | 63.85 | 91.30 |
| | $b = d_i^{\mathrm{pgt}}$ | | | ✓ | | 0.559 | 0.237 | 64.32 | 91.68 |
| | $b = d_i^{\mathrm{pgt}}$ | | ✓ | ✓ | | 0.572 | 0.242 | 64.36 | 91.37 |
| | $b = d_i^{\mathrm{pgt}}$ | ✓ | | | | 0.565 | 0.236 | 63.17 | 91.30 |
| | $b = d_i^{\mathrm{pgt}}$ | ✓ | ✓ | | | 0.550 | 0.232 | 64.49 | 91.74 |
| | $b = d_i^{\mathrm{pgt}}$ | ✓ | | ✓ | | 0.552 | 0.234 | 64.56 | 91.80 |
| | $b = d_i^{\mathrm{pgt}}$ | ✓ | ✓ | ✓ | | 0.553 | 0.235 | 64.68 | 91.88 |
| | $b = d_i^{\mathrm{pgt}}$ | ✓ | | | ✓ | 0.547 | 0.228 | 63.86 | 91.40 |
| | $b = d_i^{\mathrm{pgt}}$ | ✓ | ✓ | ✓ | ✓ | 0.542 | 0.229 | 64.76 | 91.90 |

ditionally, while the method ($b = d_i$) often degenerates the monocular depth accuracy, our method ($b = d_i^{\mathrm{pgt}}$) does not suffer from such an issue, achieving the improved monocular depth accuracy. Such a performance gain become even more apparent for both tasks when using the confidence map. Note that it is infeasible to leverage the confidence map for the method ($b = d_i$) in which the estimated monocular depth map is constantly updated during the network training. When including the cross-view consistency loss $L_{d,c}$ for monocular depth estimation, the additional performance gain was observed, validating its effectiveness on the monocular depth estimation. Though the segmentation accuracy (mIoU) was slightly worsen in some cases, it is relatively marginal. This may be due to our architecture where the two tasks share the encoder, and more advanced MTL architecture, e.g. using task-attention modules (Liu et al., 2019), would lead to performance improvement. We reserve this as future work.

## 5 CONCLUSION

This paper has presented a new MTL architecture designed for monocular depth estimation and semantic segmentation tasks. The cross-view consistency loss based on the pseudo depth labels, generated using pretrained stereo matching methods, was imposed on the prediction results of two views for resolving the mismatch problem. Intensive ablation study exhibited that it leads to a substantial performance gain in both tasks, especially achieving the best accuracy in the monocular depth estimation. Our task-specific losses can be used complementarily together with existing MTL architectures, e.g. based on task-specific attention modules (Liu et al., 2019). An intelligent combination with these approaches is expected to further improve the performance. Additionally, how to integrate recent architectures (Chen et al., 2018; Takikawa et al., 2019) designed for semantic segmentation into the MTL network would be an interesting research direction.

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

## A APPENDIX

### A.1 MORE COMPREHENSIVE EVALUATION RESULTS FOR KITTI

We provide more comprehensive results for KITTI dataset. Figure 6 shows the qualitative evaluation with existing monocular depth estimation methods on Eigen Split of the KITTI dataset. Figure 7 shows the semantic segmentation prediction results on the KITTI dataset. We also evaluated the performance with the improved ground truth depth maps made available at (Uhrig et al., 2017) for the KITTI dataset in Table 6. Our approach is state-of-the-art in monocular depth estimation compared to existing methods.

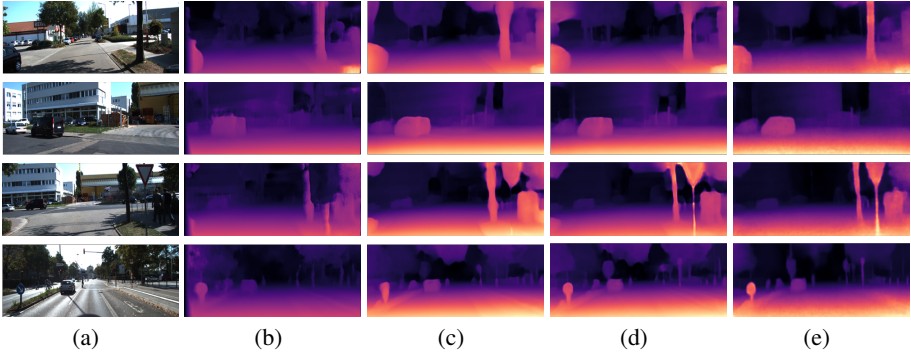

|  (a) | (b) | (c) | (d) | (e) |

Figure 6: Qualitative evaluation of monocular depth estimation on KITTI dataset. (a) Input image, (b) Monodepth (Godard et al., 2017), (c) Monodepth2 (Godard et al., 2019), (d) DepthHints (Watson et al., 2019), and (e) Ours.

Table 6: Quantitative results of monocular depth estimation using the improved ground truth depth maps made available at (Uhrig et al., 2017) for the KITTI dataset. Numbers in bold and underlined represent $1^{st}$ and $2^{nd}$ ranking, respectively. The methods used in evaluation are EPC++ (Luo et al., 2020), Monodepth2 (Godard et al., 2019), Uncertainty (Poggi et al., 2020), Packnet-SfM (Guizilini et al., 2020a), UnRectDepthNet (Kumar et al., 2020), and Choi et al. (Choi et al., 2020).

| Method | Lower is better | | | | Higher is better | | |
| --- | --- | --- | --- | --- | --- | --- | --- |
|  | Abs Rel | Sq Rel | RMSE | RMSE log | $\delta < 1.25$ | $\delta < 1.25^2$ | $\delta < 1.25^3$ |
| EPC++ | 0.120 | 0.789 | 4.755 | 0.177 | 0.856 | 0.961 | 0.987 |
| Monodepth2 | 0.090 | 0.545 | 3.942 | 0.137 | 0.914 | 0.981 | 0.994 |
| Uncertainty (Boot+Log) | 0.085 | 0.511 | 3.777 | 0.137 | 0.913 | 0.980 | 0.994 |
| Uncertainty (Boot+Self) | 0.085 | 0.510 | 3.792 | 0.135 | 0.914 | 0.981 | 0.994 |
| Uncertainty (Snap+Log) | 0.084 | 0.529 | 3.833 | 0.138 | 0.914 | 0.980 | 0.994 |
| Uncertainty (Snap+Self) | 0.086 | 0.532 | 3.858 | 0.138 | 0.912 | 0.980 | 0.994 |
| Packnet-SfM | 0.078 | 0.420 | 3.485 | 0.121 | 0.931 | 0.986 | 0.978 |
| UnRectDepthNet | 0.081 | 0.414 | 3.412 | **0.117** | 0.926 | 0.987 | 0.996 |
| Choi et al. | 0.078 | 0.357 | 3.203 | 0.120 | 0.930 | 0.987 | 0.996 |
| **Ours** | **0.076** | **0.322** | **3.094** | **0.117** | **0.933** | **0.988** | **0.997** |

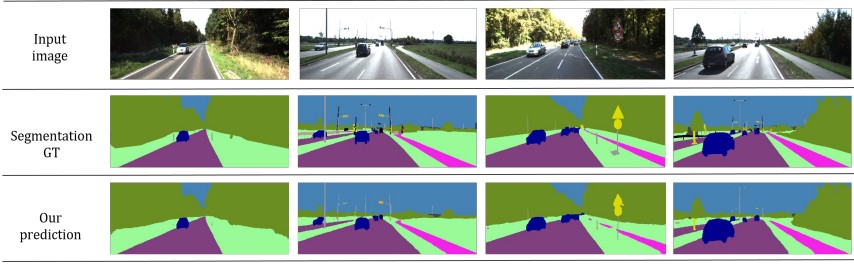

Figure 7: Qualitative results of semantic segmentation prediction on the KITTI dataset.

## A.2    MORE COMPREHENSIVE EVALUATION RESULTS FOR CITYSCAPES

We provide more comprehensive results for the Cityscapes dataset. Figure 8 and 9 show the qualitative evaluation results with existing MTL methods (Misra et al., 2016; Liu et al., 2019) for monocular depth estimation and semantic segmentation on the Cityscapes dataset. These results also support the effectiveness of our method.

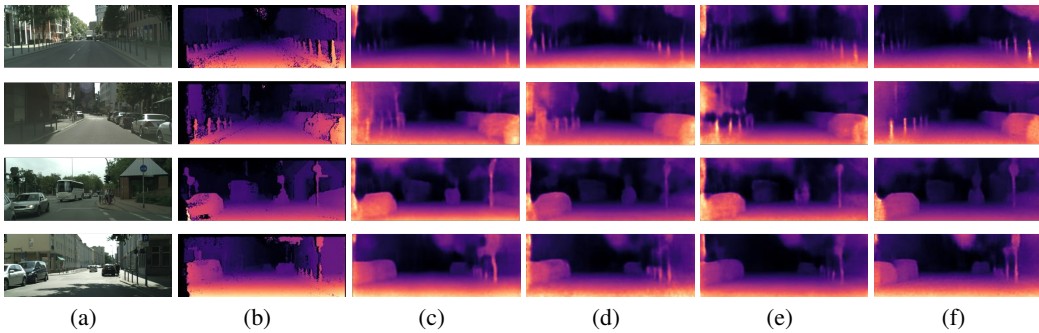

| (a) | (b) | (c) | (d) | (e) | (f) |

Figure 8: Qualitative results of monocular depth estimation on the Cityscapes dataset: (a) Input image, (b) Ground truth depth map obtained using SGM (Hirschmüller, 2008), (c) Cross-stitch (Misra et al., 2016), (d) MTAN (Liu et al., 2019), (e) Dense (Liu et al., 2019), and (f) Ours.

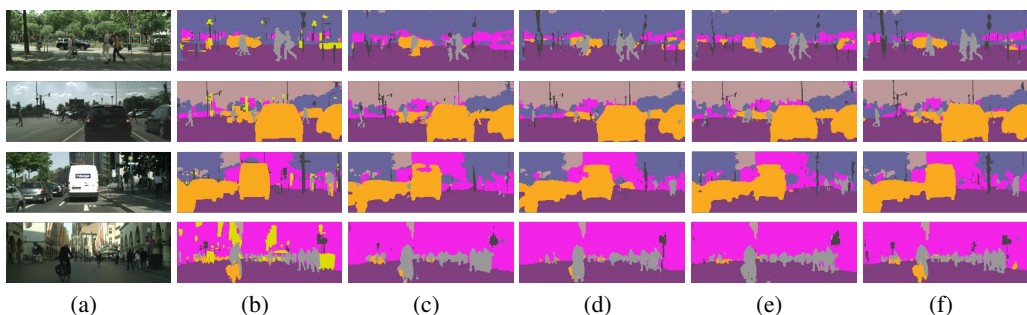

| (a) | (b) | (c) | (d) | (e) | (f) |

Figure 9: Qualitative results of semantic segmentation results on the Cityscapes dataset: (a) Input image, (b) Ground truth segmentation map, (c) MTAN (Liu et al., 2019), (d) Cross-stitch (Misra et al., 2016), (e) Dense (Liu et al., 2019), and (f) Ours.

## A.3    QUALITATIVE EVALUATION FOR ABLATION STUDY

Figure 10 and 11 and shows the qualitative evaluation for ablation study on the KITTI and Cityscape datasets, respectively. It was found that our final results are much improved compared to those of the 'Baseline' model, validating the effectiveness of the confidence map and cross-view consistency loss.

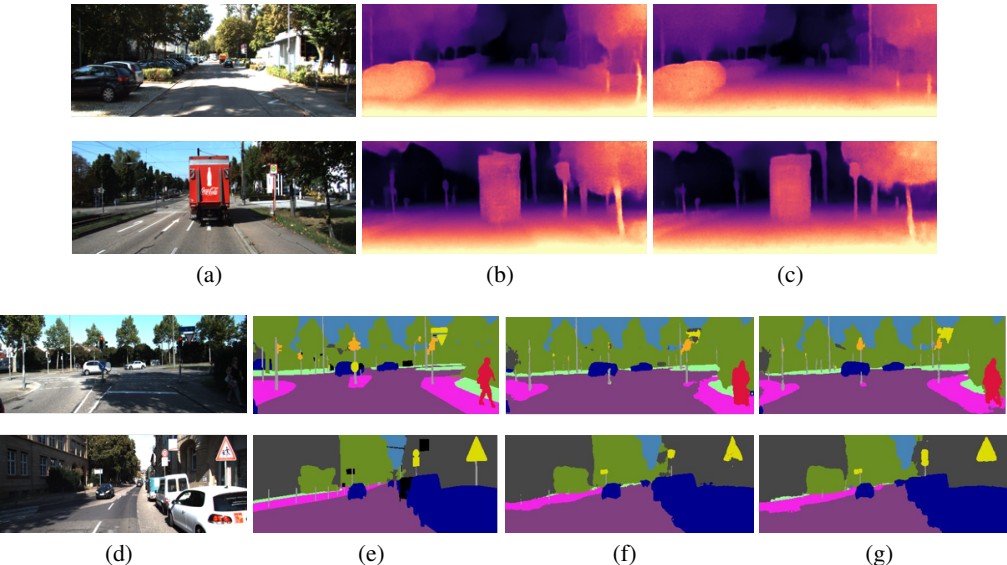

Figure 10: Qualitative evaluation for ablation study on the KITTI dataset: (a) Depth input images, (b) Depth map with 'Baseline' model, (c) Our final depth map, (d) Segmentation input images, (e) Ground truth segmentation map , (f) Segmentation map with 'Baseline' model, (g) Our final segmentation result. 'Baseline' model is our network without the confidence and cross consistency loss.

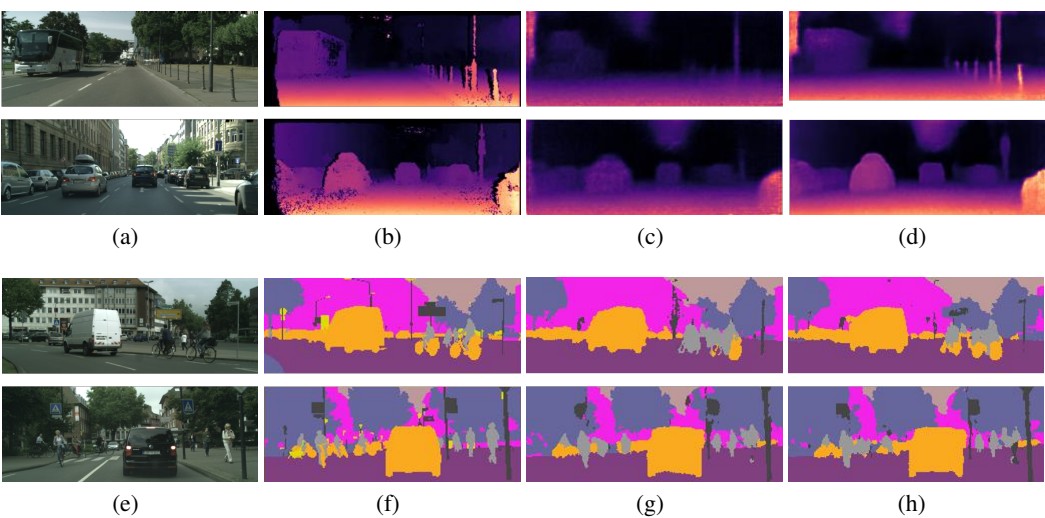

Figure 11: Qualitative evaluation for ablation study on the Cityscape dataset: (a) Depth input images, (b) Ground truth depth map, (c) Depth map with 'Baseline' model, (d) Our final depth map, (e) Segmentation input images, (f) Ground truth segmentation map, (g) Segmentation map with 'Baseline' model, (h) Our final segmentation result. 'Baseline' model is our network without the confidence and cross consistency loss.

