# OpenReview forum: "Pseudo Label-Guided Multi Task Learning for Scene Understanding"
_ICLR.cc/2021/Conference — Reject_

### Official Review · AnonReviewer2 · 2020-10-28
**too large scope for a relative limited experiments**

**Rating:** 4
**Confidence:** 5

**Review:**

This paper propose to use depth pseudo ground truth (generated with a pretrained stereo network) as augmented information to help a joint prediction network for depth and segment estimation.

Pros:
Multi task learning is an important direction to explore, and left-right consistency has shown to be very useful in depth estimation Godard et.al 2017. The extension using similiar idea to depth and semantic is reasonable, and experiments verify the effectiveness of proposed strategies.

Cons:
1) Scope:
It seems the paper works specific on the left-right warping consistency of semantic label and depth, while the major scop told in the title and introduction is about pseudo label for general multiple task learning, which is byfar not shown in the worked experiments. It needs to be adjusted.


2) Method:
The major methodology is using obtain consistency losses by warping depth and semantic with respect to stereo output. The warped loss containing 6 terms each through enumeration, are all of them useful ? Is there a lot of redundency, what happened if droping half of it. The ablation shows using concistency is useful, while the usefulness of each term and how balance between these losses has not been proven.


3) Experiments
Comparing to other SoTA algorithms, it seems for depth, the results are comparable to many existing algorithms, and for semantic it is hard to compare against other SoTA semantic algorithms such as HRNet etc.. In my opinion, MTL has two benefits either differet tasks can help the output results, another is unifying tasks into single network for more efficient inference. It might be better to also compare about the running speed and Flops for performing multiple tasks to better support the idea.

4) Writting
Overall, it is easy to follow, however the figures are too small making it hard to diagnose the difference between multiple predictions.

---

> ### Author Response · Authors · 2020-11-17
> **Response to Reviewer3**
>
> 1. Scope: It seems the paper works specific on the left-right warping consistency of semantic label and depth, while the major scope told in the title and introduction is about pseudo label for general multiple task learning, which is byfar not shown in the worked experiments. It needs to be adjusted.
>
> → I agree with your opinion about the title. We will revise the title so as to reflect the main contribution of this work. As mentioned above, the cross consistency loss used in the paper can be applied to all tasks in which stereo image pairs are available. We will conduct more experiments by applying it to all tasks.
>
> 2. Method: The major methodology is using obtain consistency losses by warping depth and semantic with respect to stereo output. The warped loss containing 6 terms each through enumeration, are all of them useful ? Is there a lot of redundency, what happened if droping half of it. The ablation shows using concistency is useful, while the usefulness of each term and how balance between these losses has not been proven.
>
> → Thank you for your suggestion. The results of dropping some losses were attached to the ablation study, and hyper-parameters for six losses are described in experiments. As you suggested, we will investigate the usefulness of each term in more details.
>
> 3. Experiments Comparing to other SoTA algorithms, it seems for depth, the results are comparable to many existing algorithms, and for semantic it is hard to compare against other SoTA semantic algorithms such as HRNet etc.. In my opinion, MTL has two benefits either differet tasks can help the output results, another is unifying tasks into single network for more efficient inference. It might be better to also compare about the running speed and Flops for performing multiple tasks to better support the idea.
>
> → While SoTA monocular depth estimation methods usually rely on simple architectures according to monocular depth estimation literatures, the SoTA segmentation methods are based on very complicated architectures. We guess our segmentation performance is not as good as SoTA segmentation algorithms since our multi-task learning architecture rely on the simple encoder-decoder architecture. As you suggested, we will compare the runtime and Flops for performing multiple tasks.
>
> 4. Writting Overall, it is easy to follow, however the figures are too small making it hard to diagnose the difference between multiple predictions.
>
> → Thank you for your suggestion. We had to put the figures small due to a page limit. Instead, more results with a relatively large size are provided in the Appendix.

---

### Official Review · AnonReviewer4 · 2020-10-28
**Several concerns on the problem setting and experimental evaluation**

**Rating:** 4
**Confidence:** 4

**Review:**

The paper presents a joint learning strategy for simultaneous semantic segmentation and monocular depth estimation. The main idea is to exploit stereo pairs in training and introduce pseudo-depth label estimated from pre-trained stereo-matching networks. Given the pseudo-depth with confidence estimation, the method proposes a cross-view consistency loss for both depth and semantic predictions, which augments the standard segmentation loss. The proposed method is evaluated on KITTI and Cityscapes datasets with comparisons to prior work and ablative study.

Strengths:
- The proposed cross-view loss on semantic segmentation seems interesting and effective on two benchmarks, which improves the segmentation performance.
- The overall method achieves competitive performance on semantic segmentation and monocular depth estimation on the KITTI and Cityscapes.

Concerns:
- The contribution of this work to the multi-task learning is a bit overclaimed. The targeted problem of the paper is solely on joint semantic segmentation and monocular depth estimation. Based on the model and loss design, it is non-trivial to extend them to other scene understanding tasks.

- The problem setting in this work, which requires stereo image pair for learning network, is different from the prior work  (e.g., Liu et al 2019). The proposed method also uses a pre-trained stereo-matching networks and confidence estimation network, which essentially included additional prior information/training data. Therefore, it is not surprising to see the performance improvement over the prior work.

- While the proposed cross-view loss improves the segmentation, the overall design is quite complicated. There are many hyper-parameters in the loss functions, and it is unclear how their values would generalize to other datasets that are not road scenes. Moreover, based on the ablative study, the improvement over the noisy depth setting is marginal (Table 4 and 5). Also, it is unclear whether all those terms make significant contributes to the performance improvements, and sometimes it even hurts the performance.

- The experimental evaluation is a bit lacking in the following aspects.
  + This work only uses two road-scene datasets for evaluation, but those two datasets are quite similar to each other, and hence do not have sufficient diversity. The other work typically also use NYU-v2, which is an indoor dataset. Can the author also report their method's performance on NYU-v2?

  + The evaluation on the Cityscapes dataset seems unconvincing due to two issues: First, the depth performance in Table 3 seems very different from the prior literature, and in particular, the Abs values are much worse than the SOTA results. Secondly, it lacks comparisons with Jha et al. 2020, which achieves better performance than the results shown in Table 3.

  + The improvement from the proposed loss seems very marginal in the ablative study. Different combinations of proposed components typically give minor or mixed improvement on segmentation or depth estimation. It is unclear how effective of the confidence weighting or using multiple consistency constraints.

---

> ### Author Response · Authors · 2020-11-17
> **Response to Reviewer2**
>
> 1. The problem setting in this work, which requires stereo image pair for learning network, is different from the prior work (e.g., Liu et al 2019). The proposed method also uses a pre-trained stereo-matching networks and confidence estimation network, which essentially included additional prior information/training data. Therefore, it is not surprising to see the performance improvement over the prior work.
>
> → Your comment is right. Indeed, we attempted to show the effectiveness of the cross-consistency loss using pseudo depth labels and confidence maps. When comparing ‘MTAN’ (Liu et al 2019) of Table 3 and ‘Baseline’ in Table 5, we can see that even the baseline of using the pseudo depth labels only outperforms 'MTAN’ (Liu et al 2019). The performance gain becomes higher by leveraging the cross-consistency loss as reported in Table 5.
>
> 2. While the proposed cross-view loss improves the segmentation, the overall design is quite complicated. There are many hyper-parameters in the loss functions, and it is unclear how their values would generalize to other datasets that are not road scenes. Moreover, based on the ablative study, the improvement over the noisy depth setting is marginal (Table 4 and 5). Also, it is unclear whether all those terms make significant contributes to the performance improvements, and sometimes it even hurts the performance.
>
> → I agree that the hyper-parameter tuning may not be easy. But, we found that the proposed loss is not much sensitive to hyper-parameters in the datasets used in experiments. We will investigate the generalization capability in diverse scenes.
> As shown in the ablation study of Table 4 and 5, it was observed that when adding the depth cross consistency view loss, the monocular depth accuracy is improved over the baseline. It was also seen that when the segmentation cross consistency view loss leads to the performance improvement in the segmentation. However, under the simple multi-task architecture sharing the encoder, boosting the depth and segmentation accuracy significantly is quite challenging. We will investigate to use more sophisticate multi-task architectures provided in recent works to address this issue.
>
> 3. The experimental evaluation is a bit lacking in the following aspects.
> This work only uses two road-scene datasets for evaluation, but those two datasets are quite similar to each other, and hence do not have sufficient diversity. The other work typically also use NYU-v2, which is an indoor dataset. Can the author also report their method's performance on NYU-v2?
> The evaluation on the Cityscapes dataset seems unconvincing due to two issues: First, the depth performance in Table 3 seems very different from the prior literature, and in particular, the Abs values are much worse than the SOTA results. Secondly, it lacks comparisons with Jha et al. 2020, which achieves better performance than the results shown in Table 3.
> The improvement from the proposed loss seems very marginal in the ablative study. Different combinations of proposed components typically give minor or mixed improvement on segmentation or depth estimation. It is unclear how effective of the confidence weighting or using multiple consistency constraints.
>
> → Thank you for your valuable suggestion. Unfortunately, it is infeasible to apply the cross consistency loss to NYU-v2 dataset that provides only a single image, not stereo image pairs. We will seek various datasets for conducting experiments for ensuring the diversity, as you suggested.
> In the prior literatures, the performance in the Cityscapes dataset was usually measured with disparity maps obtained using the hand-crafted stereo matching method, semi-global matching (SGM) [Hirschmuller, 2008]. We found that the SGM disparity map used for the performance evaluation contains disparity values that are 0 or close to 0 at many parts. Since these values are meaningless, we excluded these values in the performance evaluation. Note that for a fair comparison, we measured the performance of all methods under the same setup. The code will be publicly available soon. As suggested, we will include the comparison with Jha et al. 2020.

---

### Official Review · AnonReviewer3 · 2020-10-29
**This paper presents a framework which leverages pseudo depth ground truth to train monocular depth and semantic segmentation networks.**

**Rating:** 3
**Confidence:** 5

**Review:**

The paper presents a framework to learn depth prediction and semantic segmantation jointly; the key idea lies in making use of the pseudo depth label from stereo to provide supervision and as a means to enforce cycle consistency between the left and right views of the stereo.

Reasons for scores: overall the paper is rather incremental and the idea is neither novel nor significant in my opiniont. I do not see interesting or deep insight from the paper towards the depth and semantic segmantation tasks.

Pros:
+ First, the paper is clearly written and easy to follow. The proposed framework is pretty straightforward.
+ The idea of joint learning depth and semantic segmentation is good considering their tightly coupled nature.
+ The use of cross-view consistency as a constraint is good.

Cons:
- Overall, the paper does not have much novelty in my opinion. Joint learning of depth and semantic segmentation is clearly not new, and the paper does not provide new or particular insight towards this combined learning.
- The use of pseudo label itself is nowadays quite common in the vision community.  And, the pseudo labels are used in the paper in a pretty trivial way in my opinion.
-  The cross-view consistency across two views in a stereo setup is not new neither. It has been intensively used in the monocular depth estimation. In addition, this constraint is applicable to any individual task and does not seem to fit into the multi-task learning context, which is the main focus of this paper. I would expect specific insights in making use of pesudo labels towards solving the depth and semantics predictions; otherwise, any other tasks such as moving objects segmentation
- The current title is too general, so much so that the main arguments made by the paper are not reflected in the title; I believe that the left-right consistency brought about by the pseudo ground truth depth is the main claim of the paper.

---

> ### Author Response · Authors · 2020-11-17
> **Response to Reviewer1**
>
> 1. Overall, the paper does not have much novelty in my opinion. Joint learning of depth and semantic segmentation is clearly not new, and the paper does not provide new or particular insight towards this combined learning.
> The use of pseudo label itself is nowadays quite common in the vision community. And, the pseudo labels are used in the paper in a pretty trivial way in my opinion.
> 2.The cross-view consistency across two views in a stereo setup is not new neither. It has been intensively used in the monocular depth estimation. In addition, this constraint is applicable to any individual task and does not seem to fit into the multi-task learning context, which is the main focus of this paper. I would expect specific insights in making use of pseudo labels towards solving the depth and semantics predictions; otherwise, any other tasks such as moving objects segmentation
>
> → Thank you for your analysis and suggestions. In our humble opinion, while existing works [Godard et al., 2017; 2019; Watson et al., 2019; Chen et al., 2019] for imposing the cross-view consistency across two views use predicted disparity maps, our method attempts to impose the cross-view consistency by making use of pseudo disparity maps and their associated confidences. We showed the effectiveness of the cross-view consistency based on the pseudo label and its confidence through the ablation study in Table 4. We could see that when the cross-consistency loss was applied by using (incomplete) predicted disparity, it does not improve performance. We believe this is a difference from the existing papers [Godard et al., 2017; 2019; Watson et al., 2019; Chen et al., 2019]. Nevertheless, we will continue to investigate the applicability of using pseudo labels in the depth and semantics predictions, as you suggested.
>
> 3. The current title is too general, so much so that the main arguments made by the paper are not reflected in the title; I believe that the left-right consistency brought about by the pseudo ground truth depth is the main claim of the paper.
>
> → I agree with your opinion about the title. We will revise the title so as to reflect the main contribution of this work. As mentioned above, the cross consistency loss used in the paper can be applied to all tasks in which stereo image pairs are available. We will conduct more experiments by applying it to all tasks.

---

### Decision · Program_Chairs · 2021-01-07
**Final Decision**

**Decision:**

Reject

**Comment:**

All reviewers agree that the paper overclaims its contributions both in the main text and in the title, and given also the limited novelty  and scope it is not suggested for publication.